# The Impact of Body Mass Composition on Outcome in Multiple Traumatized Patients—Results from the Fourth Thoracic and Third Lumbar Vertebrae: A Single-Center Retrospective Observational Study

**DOI:** 10.3390/jcm12072520

**Published:** 2023-03-27

**Authors:** Esref Belger, Daniel Truhn, Christian David Weber, Ulf Peter Neumann, Frank Hildebrand, Klemens Horst

**Affiliations:** 1Department of Orthopedics, Trauma and Reconstructive Surgery, RWTH Aachen University Hospital, 52074 Aachen, Germany; 2Department of Surgery and Transplantation, RWTH Aachen University Hospital, 52074 Aachen, Germany; 3Department of Diagnostic and Interventional Radiology, RWTH Aachen University Hospital, 52074 Aachen, Germany

**Keywords:** multiple trauma, polytrauma, body mass composition, outcome, complication, mortality

## Abstract

Background: Body mass composition (BC) was shown to correlate with outcome in patients after surgery and minor trauma. As BC is assessed using computed tomography (CT) and routinely applied in multiple trauma (MT), this study will help to analyze whether BC variables also correlate with outcome in trauma patients. Materials and Methods: Inclusion criteria were MT (Injury Severity Score (ISS) > 15) and whole-body CT (WBCT) scan on admission. Muscle and fat tissue were assessed at the level of the fourth thoracic vertebra (T4) and the third lumbar vertebra (L3) using Slice-O-matic software, version 5.0 (Tomovision, Montreal, QC, Canada). Univariate and multivariate regression models were used with regard to outcome parameters such as duration of ventilation, hospital stay, local (i.e., pneumonia, wound infection) and systemic (i.e., MODS, SIRS) complications, and mortality. Results: 297 patients were included. BC correlated with both the development and severity of complications. Skeletal muscle index (SMI) and subcutaneous adipose tissue index (SATI) at both T4 and L3 correlated positively with the occurrence of systemic infections. Local infections positively correlated with SMI at T4. Low muscle mass and high visceral adipose tissue (VAT) predicted the severity of systemic and local complications. Muscle tissue markers at both T4 and L3 predicted the severity of complications in roughly the same way. Moreover, higher muscle mass at the L3 level was significantly associated with higher overall survival, while SATI at the T4 level correlated positively with hospital stay, length of stay in the ICU, and duration of ventilation. Conclusions: A lower muscle mass and a high adipose tissue index are associated with a poor outcome in MT. For the first time, it was shown that BC at the fourth thoracic vertebra is associated with comparable results to those found at the third lumbar level.

## 1. Introduction

Despite significant improvements in prevention and therapy, mortality in multiple traumatized (MT) patients remains high. In addition to injury-related death at the scene of the accident, high mortality rates result from complications such as secondary brain damage, sepsis, and multiple organ dysfunction [1]. Analysis of body mass composition (BC) has previously been described to predict outcome in various patient cohorts; in patients with cardiovascular or chronic obstructive pulmonary diseases, high muscle mass correlated well with improved survival [2,3]. The loss of skeletal muscle mass and function (sarcopenia) is associated with poor outcome in surgical patients and after minor trauma [4]. In addition to longer hospital stays and post-operative complications, a higher mortality rate was described in the latter cohorts [2,4,5]. Against this background, BC analysis at the third lumbar vertebra (L3) was validated to correlate well with the whole-body skeletal muscle mass [6,7].

However, the literature regarding BC reporting on outcome parameters in severely injured patients is rare. As computed tomography (CT) scans are one of the most accurate methods for tissue-related quantification of BC and as it is performed routinely in MT patients, BC variables at the L3 level might be analyzed easily with regard to patient outcome [8]. However, due to an isolated traumatic impact to the upper body part (e.g., thoracic injury), CT data for BC analysis might only be available for the thoracic spine and not for the lumbar spine because of the absence of an insult at the lumbar level. Therefore, we hypothesized that BC at the thoracic level might offer comparable information to patient status and thus help to predict outcome in a comparable way to the more validated L3 level. Although the technique was described on the level of the fourth thoracic vertebra (T4) in patients with advanced lung disease by Mathur et al. [9], no data exist regarding the trauma population.

Thus, the main goals of the present study were to (a) determine whether there is a correlation between CT-based BC indices and the development and severity of local (i.e., pneumonia, wound infection), systemic (i.e., SIRS, MODS), as well as intraoperative complications and their influence on patient clinical course and outcome and (b) to determine whether BC on a thoracic level provides comparable information to that gained from the validated lumbar level with regard to risk stratification in MT patients.

## 2. Materials and Methods

### 2.1. Study Design

Patients aged ≥18 years diagnosed with multiple trauma (ISS > 15), admitted to RWTH Aachen University hospital between January 2010 and December 2015 were included into the single-center retrospective observational study. The RWTH Aachen University hospital is a level 1 trauma center and has more than 1400 hospital beds, and 165 of these are intensive care unit (ICU) beds [10]. All patients routinely received a whole-body CT (WBCT) scan on the day of accident as part of the trauma protocol. Patients were excluded if their CT scans did not show the thoracic or abdominal wall or if the CT scan showed too many artefacts so that BC variables could not be definitely determined.

Primary endpoints were the development and the severity of local and systemic complications after MT. Secondary endpoints were defined as the length of ventilation (hours), the length of ICU and total hospital stay (days), and mortality.

### 2.2. The Comprehensive Complication Index (CCI) Score

CCI is a morbidity score that measures the overall extent of all complications [11]. The square root of the sum of all complications graded by the Clavien–Dindo (CD) classification divided by 2 provides a single morbidity score from 0 (no complications) to 100 (death) [11,12].

### 2.3. Injury Severity Score (ISS)

ISS is a medical score to assess the severity of injury, which correlates with mortality and morbidity. Injury Severity Scores range from 1 to 75. An ISS higher than 15 is defined as a major trauma [13].

### 2.4. Definition of Complications

The occurrence of complications and death was documented during the patient clinical course. Complications were divided into local (i.e., pneumonia, infection, urinary tract infection, infection of the wound, wound healing disorder, thrombosis, compartment syndrome, pulmonary embolism, apoplexy), systemic (i.e., systemic inflammatory response syndrome (SIRS), sepsis, multiorgan dysfunction syndrome (MODS)), and surgically acquired (i.e., nerve damage, vascular damage, bleeding) complications.

### 2.5. The Clavien–Dindo (CD) Classification

CD classification is a widely accepted classification system consisting of seven grades to define complications [14,15]. The term minor is used for complications with a CD score of I or II. The term major is used for complications with a CD score of III or higher [14]. Grades I and II indicate minor deviations from the normal clinical course, and IIIa complications require surgical intervention without anesthesia. Grade IV complications are life-threatening, and those of grade V result in death [16]. Even in complex scenarios, a significant interrater reliability (>90%) was proven. Thus, the CD is also feasible for physicians who are less experienced [14,16]. Complication severity affects further clinical course, so CD classification was determined regardless of local or systemic complications.

### 2.6. BC Analysis Using CT

CT scans performed during the initial assessment of MT patients at the time of hospital admission were analyzed by two experienced researchers (E.B., K.H.). The software used was Slice-O-matic, version 5.0 (Tomovision, Montreal, QC, Canada).

The third lumbar vertebra was used as a landmark to calculate the tissue cross-sectional area in square centimeters. Furthermore, the skeletal muscle (SM), visceral adipose tissue (VAT), and subcutaneous adipose tissue (SAT) were identified and quantified on the CT scans by using predetermined Hounsfield units (SM: (−29)—150 HU, VAT: (−150)—(−50) HU, SAT: (−190)—(−30) HU) (Figure 1 and Figure 2). After that, the variables were corrected by considering the height of the patients. This way the body composition indices skeletal muscle index (SMI), subcutaneous adipose tissue (SATI), and visceral adipose tissue (VATI) in square centimeters per square meter were calculated [17]. Additionally, skeletal muscle radiation attenuation (SMRA) was determined, which is a radiological characteristic, a low SMRA is suggestive of intramuscular adipose tissue infiltration and poor “quality” skeletal muscle [18]. The same process was also performed for the fourth thoracic vertebra (T4), with the difference that no visceral adipose tissue (VAT) needed to be analyzed.

### 2.7. Statistical Analysis

IBM SPSS Statistics version 26.0 (IBM Corp., Armonk, NY, USA) was used to perform all statistical analyses. 

First, demographic and clinical data as well as occurrence of overall complications and their severity were correlated to BC indices using Pearson correlation, Kendall Tau, Spearman tests, and eta-coefficient according to the variable characteristics (i.e., binary, continuous).

Logistic regression analysis (LRA): Logistic regression was used for binary dependent variables, while for continuous variables; a linear regression model was applied. A *p*-value of <0.05 was considered statistically significant. For the outcome of mortality, the Cox regression model was used to determine the association between each body composition variable and the clinical outcome.

Multivariable regression analysis (MRA): Statistically significant variables from LRA with a *p*-value of <0.05 were used to determine an association between the BC variables and the occurrence and severity of both systemic and local complications. Multivariable Cox regression analysis was used to determine the association between overall survival and the different study variables. A *p*-value of <0.05 was considered statistically significant.

## 3. Results

### 3.1. Demographic Data

Overall, 505 MT patients were identified. Out of these, 297 met the inclusion criteria (Figure 3). Detailed data regarding demographic and trauma pattern are depicted in Table 1, Table 2 and Table 3. The average hospital stay was 23 days (SD 26.14), while the median stay in the ICU was 15 days (SD 23.6). The average number of complications that occurred during hospital stay was three (SD 3) (Table 2). Detailed information regarding frequency and severity of complications are depicted in Table 3 and Table 4. In total, 68 patients (22.9%) died after admission or during the further clinical course (Table 1).

### 3.2. Correlations

The cohort showed that SMRA and SMI decreased, while VATI and SATI increased with increasing age (Appendix A). Furthermore, T4 SATI but not L3 SATI correlated positively with hospital stay, ICU stay, and duration of ventilation (Appendix A). L3 SMRA, L3 SMI, T4 SMRA, and T4 SMI were negatively correlated with CD score, while L3 VATI showed a positive correlation with CD score (Appendix A).

### 3.3. Linear Regression

Low muscle tissue markers (SMRA and SMI) at both the T4 and L3 level and a higher L3 VATI were associated with a higher CD score (Appendix A). SMRA and SMI assessed at the fourth thoracic vertebra and the third lumbar vertebra both predicted the severity of overall complications in roughly the same way. In detail, increased SMI and SATI at both the T4 and L3 level and increased L3 SMRA were significantly associated with the occurrence of systemic infections (Appendix A). Local infections showed an association only with increased T4 SMI (Appendix A). A detailed analysis of all local complications revealed that pneumonia was associated with increased T4 SMI. Moreover, a higher T4 SATI showed an association with a longer hospital stay, a longer ICU stay, and a longer duration of ventilation (Appendix A). Increased L3 SMRA and L3 SMI showed a statistically significant association with regard to overall survival. In contrast, at the thoracic level, only increased SMRA showed a significant association with overall survival (Table 5). A high SMRA at both the T4 and L3 level and a high L3 SMI were associated with a lower probability of dying. Comparable to L3, it was able to predict overall survival by assessing SMRA at T4 (Table 5).

### 3.4. Multivariable Analysis

The multivariate analysis contained the variables age, ISS, CCI and all the variables that were significant in univariate analysis.

Muscle tissue markers (SMRA and SMI) assessed at the fourth thoracic vertebra and the third lumbar vertebra both predicted the overall severity of complications in roughly the same way (Appendix A). However, the results indicated that a higher CD score and thus the probability for an adverse clinical course including additional therapeutic procedures (i.e., antibiotics, dialysis, intensive care treatment) was significantly associated with reduced T4 SMI and L3 SMI (Appendix A). Regarding local complications, the multivariable analysis presented a significant association with increased T4 SMI (Appendix A). Increased SMI and SATI at both L3 and T4 level showed an association with occurrence of systemic complications. Furthermore, the association between a higher T4 SATI and a longer hospital stay, a longer ICU stay, and a longer duration of ventilation was also shown in multivariable analysis (Appendix A).

The multivariate Cox regression analysis contained the variables age, ISS, CCI, occurrence of overall complications, local complications, and all the variables that were significant in the univariate analysis. This showed an association of increased L3 SMI, hospital stay, and ICU stay with overall survival (Table 5).

## 4. Discussion

MT remains one of the leading causes of death, especially in the young population. [19]. Dreadful complications occur frequently during the further clinical course and are often responsible for death after patients survive the initial insult [20]. The early identification of patients at risk therefore is crucial to adapt therapy strategies, and BC analysis might offer a new tool to predict the development of complications and improve outcomes in this special population. The findings from the current study can be summed up as follows:Increased SMI and SATI at both the T4 and L3 level were significantly associated with the occurrence of systemic infections. Local infections showed an association with increased T4 SMI. Information from both levels provides complementary information.Severity of posttraumatic complications negatively correlated with SMI at both, the L3 and T4 level. BC variables at both levels predicted the severity of overall complications in roughly the same way.SATI at T4 allowed us to make an estimation about the duration of ventilation, length of stay in the ICU, and total hospital stay.Increased SMI at the L3 level significantly correlated with overall survival.

### 4.1. Development and Severity of Complications

Previous studies in different surgical populations have reported on the impact of BC with regard to patient outcome. In oncologic patients, low SMI correlated with increased morbidity and mortality [21]. Additionally, data from patients receiving liver transplantation showed that myosteatosis is an independent predictor of major surgical complications [22]. Patients with high visceral adipose tissue [23] or high intramuscular adipose tissue were found to be associated with postoperative complications [24]. As the literature regarding the trauma population is sparse, the present study revealed comparable findings to for this cohort. Recently, Poros et al. investigated whether fat or muscle tissue correlates with key elements of pre-hospital and clinical care in an adult population with severe trauma [25]. In contrast to Poros et al., who did not find any prevalence of sepsis with regard to fat/muscle tissue markers, the present study demonstrated a positive correlation between SMI and SATI at both L3 and T4 level and the occurrence of systemic complications. This might be due to the inclusion criteria of the present study. In contrast to Poros et al., who did not report on the overall injury severity of their patient cohort, this is the first study with a well-defined MT population presented with a mean ISS of 25 points. Data published by Andruzskow et al. showed obesity as an independent risk factor for complications in severely traumatized patients [26]. A high amount of fat tissue contributes to inflammation which in turn cause a deranged post-traumatic immunologic reaction promoting the development of complications [27]. This coherence can also be drawn from the present study. Furthermore, it is well-known that severe tissue injury is associated with an increased concentration of inflammatory parameters, which positively contribute to the development of systemic inflammation in MT patients [28,29]. Systemic inflammatory response syndrome and multiple organ failure may occur [30,31,32]. Thus, the observed findings extend the current literature and offer a new predictive parameter with regard to the development of systemic complications in MT patients. Additionally, SMI on the level of the fourth thoracic vertebra was positively associated with the development of local complications, of which pneumonia was the most prominent. As reported from other non-trauma cohorts, the present data confirm a positive correlation between BC variables and the occurrence of pneumonia in a trauma population [33]. Furthermore, the present study demonstrated a negative correlation between SMI at both L3 and T4 and the severity of complications. The lower the SMI, the more severe the observed complications. As muscle mass represents the physical status of a patient, this observation could be indicative of a reduced physical condition and thus less activity, a higher degree of frailty, and increased susceptibility to the development of complications in trauma patients as shown in other non-trauma populations before [34].

While Poros et al. did not find any association between BC variables and the course of treatment, in particular the duration of mechanical ventilation, ICU length of stay, and neurologic outcome, the present study revealed that SATI at T4 allowed us to make an estimation about hospital stay, length of stay in the ICU, and duration of ventilation. These contrasting results might be explained by the divergent inclusion criteria. In contrast to Poros et al., the current study reports for the first time on a well-defined population of MT patients. A significantly longer ventilation time and ICU stay as well as higher mortality are known to correlate well with injury severity, which might explain the differences between our study and the results of Poros et al. [35,36]. Additionally, Poros et al. included patients regardless of age [25], while the present work excluded children and adolescents. Furthermore, different analytic techniques regarding the measurement of BC were applied. Poros et al. focused on BC variables on the level of the third lumbar vertebra only and used different Hounsfield unit thresholds than applied in the present study. However, diverse other studies support the application of the Hounsfield unit thresholds used in the present study, which supports these findings in MT patients [6,33].

To sum up, using the initial CT scan and taking BC variables into account might help to predict the occurrence and severity of complications which significantly influence the further clinical course of severely injured patients. Moreover, BC variables help to estimate the duration of ventilation as well as length of stay in the ICU and hospital and thus might allow for optimizing treatment procedures and strategies.

### 4.2. L3 vs. T4

As described, the analysis of BC at the third lumbar vertebra was validated to correlate well with the whole-body skeletal muscle mass [6,7]. As MT patients regularly sustain an isolated impact to the upper body region (e.g., thoracic trauma) CT data of the lumbar spine might not be available. Therefore, it is clinically relevant to investigate if BC variables measured at the thoracic level also correlate with the clinical course of the patient or predict outcome in trauma patients. The present study revealed a negative correlation between both SMRA and SMI, and the severity of emerging complications identified by the CD score at both levels (L3 and T4). Moreover, SATI at T4 allowed us to make an estimation about the duration of ventilation, length of stay in the ICU, and total hospital stay. In addition to previous studies from other patient cohorts that already demonstrated a correlation between the thoracic muscle cross sectional area at T4 and the whole muscle volume [9] as well as outcomes in patients with advanced lung disease [37], the present investigation also found a relevant correlation of BC at both the T4 and L3 levels with regard to outcome parameters in trauma patients. Especially the new information that BC variables assessed at the T4 level are partially complementary to those assessed at the L3 level and that BC variables at T4 also allow additional statements about duration of ventilation, length of stay in the ICU, and total hospital stay might be a helpful alternative to predict the further clinical course and adapt therapeutic regimens.

### 4.3. Hospital Stay and Mortality

In addition to a positive correlation with ICU and total hospital stay, mortality is also associated with BC variables [22,38]. However, previous studies reporting on these findings mainly focused on patients with sarcopenia or myosteatosis. With regard to the trauma population, Moisey et al. showed an association between low muscle mass and mortality in older (≥65 years old) patients [39]. However, the authors only reported on patients with minor trauma, not on severely injured patients. Although Poros et al. reported on younger trauma patients, the authors did not analyze a possible correlation between BC and mortality [25]. As discussed earlier, data from this study might not adequately represent MT, which is further expressed by a remarkable low mortality rate of only 10% [25]. Data from the current study now reveal that reduced muscle mass is also positively correlated to mortality in a well-characterized population of younger MT patients [39,40]. Stassen et al. described a similar correlation between low SMI and 30-day mortality in a polytraumatized cohort of older patients (≥80 years old) [41]. It might be suggested that low muscle mass represents a general reduced physical state, contributing to the development of severe complications that in turn are responsible for the observed rather high mortality rate of 22.9%. Additionally, the present study revealed that MT patients with high subcutaneous fat tissue were in need of ventilation for a longer period and had a prolonged stay in the ICU and in hospital. These findings are in line with previously published observations on obese trauma patients [42,43,44]. As an underlying cause, Covarrubias et al. reported on obesity being responsible for decreased chest wall compliance, decreased lung volume, increased oxygen consumption, increased respiratory rate, mild hypoxemia, and increased airway resistance [44]. Additionally, data presented by Andruszkow et al. revealed that obesity is an independent risk factor for a complicated course in severely traumatized patients with an associated longer ICU and in-hospital stay [26]. In contrast to data presented by Bouillanne et al., who reported on reduced morbidity and mortality in elderly patients (≥65 years old) with increased fat mass [40], the present study showed no association between fat mass and overall survival. This might be due to the fact that Bouillane et al. reported on a cohort aged 83.8 ± 7.7 years old, presenting an average BMI of 22.7 ± 4.0 in men and 23.3 ± 4.9 in women [40], which is clearly lower than that observed in the current study (26.02 ± 4.22). Although most patients in the Bouillane study participated in rehabilitation after fracture (34%), a huge percentage suffered from other non-traumatic diagnoses such as neurologic disease (26%), postinfectious disease (13%), cardiovascular disease (CVD; 9%), and other medical diseases (18%) [40] that might have further biased the results in comparison to MT patients. In summary, the current study gives new and important information regarding the relevance of different BC variables in view of hospitalization and mortality of MT patients and thus might not only help to identify patients at risk but also will help to optimize the clinical course of this special subpopulation. 

### 4.4. Strengths and Limitations

Limited data is available on the relevance of BC for the clinical course and outcome of MT patients. Most studies focus on sarcopenic patients or those who suffer from cancer with mortality as the only outcome parameter. BC analysis to determine the association with occurrence and severity of complications and the impact on the clinical course in a well-defined population of MT patients has not been performed before. This study therefore gives new and important insights into a considerable number of MT patients. Moreover, the present study focused on T4 and L3 to identify the body composition variables and therefore showed that measuring BC at T4 can adequately be performed in comparison to L3, which has already been established as the gold standard.

However, MT patients are characterized by a high heterogeneity (injury distribution, age, gender, premedical condition), which might contribute to a bias compared to other populations. Additionally, the present study includes patients with MT who were admitted to a level 1 trauma center and a hospital of supramaximal care. Thus, the observed correlations could not be referable to patients who are threatened in hospitals that are not comparable. Furthermore, positioning of all patients in the CT scanner varies, which causes variable axial orientations of the body in the CT scanner. This could lead to variations in segmentation and measurement of body composition. Thus, data regarding BC variables should always be interpreted with great care.

## 5. Outlook

Weston et al. showed a fully automated algorithm for segmenting abdominal CT images [45]. As described, on the basis of SATI at T4, duration of ventilation, length of ICU and total hospital stay might be predicted and thus help to optimize treatment strategies and patient management. Moreover, various other clinically relevant variables might be associated with BC and should be investigated. Yet, threshold-automated programs have been shown not to reliably discriminate between the density of fat tissue and gastrointestinal organs [40]. Further studies therefore need to focus on the improvement of automated programs and the use of artificial intelligence. Moreover, patients with MT have a long bed rest time. The fact that bed-ridden patients lose muscle mass should be considered regarding the time point of analysis. A prospective study would help to evaluate the development of body composition after multiple trauma by analyzing CT scans performed over the course of the hospital stay.

## 6. Conclusions

In conclusion, this study shows that the occurrence and severity of complications, ventilation time, stay in the ICU and in hospital, as well as overall survival in MT patients can be predicted using CT scan-based BC variables at both the L3 and T4 levels. BC variables assessed at the T4 and L3 levels provide partial complementary information, but especially T4 SATI is suitable to make a statement about the duration of ventilation, length of stay in the ICU, and total hospital stay. Furthermore, SMI can be used to make a statement about overall survival and to predict the occurrence of systemic or local complications and the severity of complications. This might help to optimize treatment strategies and resource management in MT patients. Therefore, new methods should be developed to facilitate and accelerate routinely the assessment of BC on admission.

## Figures and Tables

**Figure 1 jcm-12-02520-f001:**
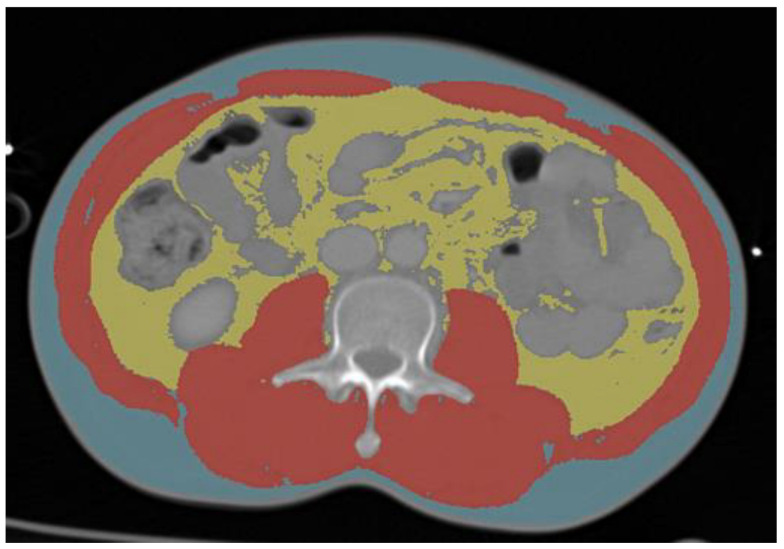
Segmentation at L3. Yellow: L3 visceral adipose tissue; red: L3 skeletal muscle tissue; blue: L3 subcutaneous adipose tissue.

**Figure 2 jcm-12-02520-f002:**
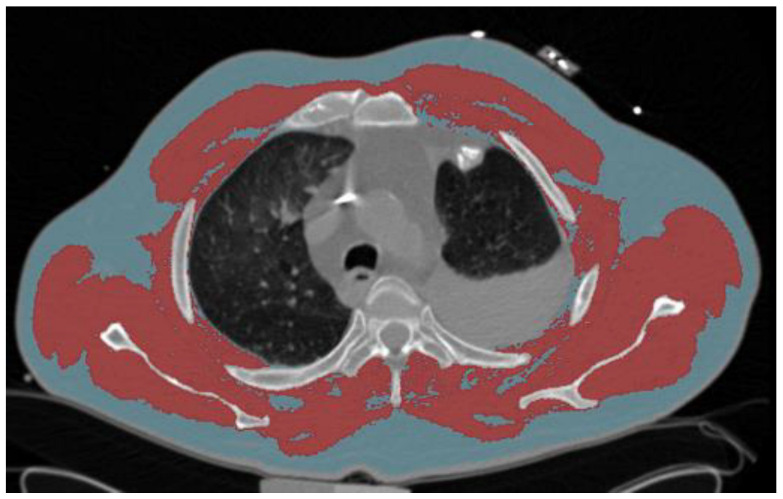
Segmentation at T4. Red: T4 skeletal muscle tissue; blue: T4 subcutaneous adipose tissue.

**Figure 3 jcm-12-02520-f003:**
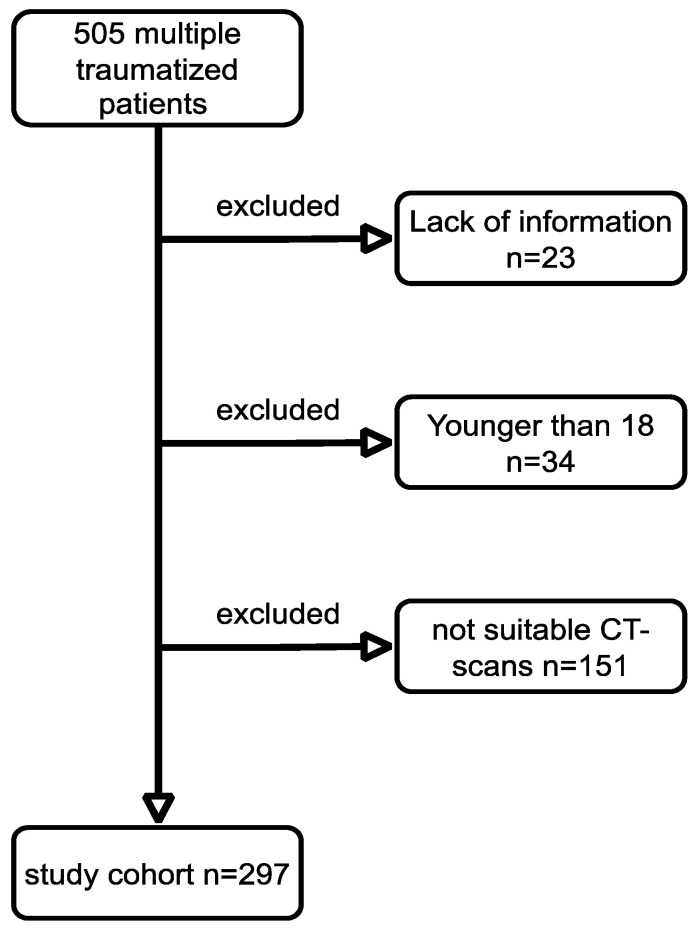
Flowchart showing the formation of the cohort.

**Table 1 jcm-12-02520-t001:** Demographic data.

Demography	Frequency	Mean	Minimum	Maximum	SD	Variance
**Total (n)**	297					
**Male (n)**	220 (74.07%)					
**Death (n)**	68 (22.9%)					
**Age (y)**		49.01	18	93	19.99	399.39
**BMI**		26.02	16.89	55.80	4.22	17.81
**Height (cm)**		176.28	150	202	8.792	77.30
**Weight (kg)**		81.18	38	165	15.69	246.08

BMI: body mass index, SD: standard deviation.

**Table 2 jcm-12-02520-t002:** Type of injury and trauma etiology.

Type of Injury	Frequency
**Neurotrauma**	165 (55.56%)
**Chest trauma**	193 (64.98%)
**Abdominal trauma**	105 (35.35%)
**Pelvic trauma**	84 (28.28%)
**Trauma Etiology**	**Frequency**
**Car/Truck accident**	59
**Motorcycle accident**	54
**Bicycle accident**	16
**Fall < 3 m**	39
**Fall > 3 m**	53
**Hit pedestrians**	30
**Burn and blast trauma**	8
**Other or unknown**	38

SD: standard deviation.

**Table 3 jcm-12-02520-t003:** Clinical course paramters.

Clinical Course	Mean	Minimum	Maximum	SD	Variance
**Frequency of complication**	3.05	0	20	3.169	10.041
**Hospital stay (d)**	23.15	1	263	26.144	683.503
**Length of stay in the ICU (d)**	15.03	1	263	23.595	556.715
**Survival days**	283.30	1	365	150.357	22,607.373
**Ventilation duration (h)**	222.72	0	5496	443.52	8196.432
**GCS**	9.62	3	15	5.139	26.41
**CD**	3.52	0	7	2.372	5.629
**CCI**	51.0178	0	100	51.43245	2645.297
**ISS**	25.17	8	75	8.341	69.575
**AIS skull and cervical spine**	2.05	0	5	1.862	3.468
**AIS thorax**	1,98	0	5	1.529	2.338
**AIS abdomen**	0.92	0	5	1.257	1.581
**AIS extremities**	1.66	0	5	1.443	2.081
**AIS soft tissue**	0.46	0	6	0.912	0.832
**RR systolic on admission**	128.1	0	233	33.042	1091.759
**RR diastolic on admission**	71.08	0	130	19.574	383.132

ICU: intensive care unit; CD: Clavien–Dindo score; CCI: Comprehensive Complication Index; ISS: Injury Severity Score; SD: standard deviation.

**Table 4 jcm-12-02520-t004:** Detailed overview of complications.

Complication	Frequency	Percentage
**Intraoperative bleeding**	8	2.7%
**Intraoperative damage to nerves or vessels**	0	0%
**SIRS**	48	16.2%
**ARDS**	52	17.5%
**MODS**	3	1%
**Myocardial infarction**	91	30.6%
**Apoplexy**	7	2.4%
**Thrombosis**	6	2%
**Pulmonary embolism**	5	1.7%
**Compartment syndrome**	7	2.4%
**Wound infection**	31	10.4%
**Wound healing disorder**	16	5.4%
**UTI**	20	6.7%
**Pseudarthrosis**	5	1.7%
**Pneumonia**	62	20.9%
**Hemothorax**	30	10.1%
**Pneumothorax**	78	26.3%
**Tension pneumothorax**	12	4%

SIRS: systemic inflammatory response syndrome; ARDS: acute respiratory distress syndrome; MODS: multiple organ dysfunction syndrome; UTI: urinary tract infection.

**Table 5 jcm-12-02520-t005:** Regression models of BC parameters with regard to overall survival.

	Univariable			Multivariable		
	Hazard Ratio	95% CI	*p*	Hazard Ratio	95% CI	*p*
**L3 SMRA**	**0.966**	**0.947–0.986**	**0.001**	0.982	0.952–1.013	0.247
**L3 VATI**	1.002	1.000–1.005	0.092			
**L3 SMI**	**0.990**	**0.981–0.999**	**0.022**	**0.988**	**0.978–0.998**	**0.022**
**L3 SATI**	0.999	0.996–1.001	0.268			
**T4 SMRA**	**0.962**	**0.934–0.990**	**0.009**	1.005	0.967–1.045	0.790
**T4 SMI**	0.997	0.992–1.002	0.209			
**T4 SATI**	1.000	0.997–1.002	0.772			
**Age**	**1.029**	**1.016–1.042**	**0.000**	**1.033**	**1.021–1.046**	**0.000**
**ISS**	**1.042**	**1.019–1.064**	**0.000**	**1.058**	**1.029–1.087**	**0.000**
**Clavien-Dindo**	**6.224**	**3.808–10.175**	**0.000**	207.429	0.000–9.521 × 10^9^	0.533
**CCI**	**1.005**	**1.003–1.006**	**0.000**	**1.009**	**1.007–1.011**	**0.000**
**Occurrence of complication**	**0.433**	**0.266–0.707**	**0.001**	**0.437**	**0.256–0.746**	**0.002**
**Local complications**	**0.289**	**0.152–0.552**	**0.000**	**0.163**	**0.069–0.386**	**0.000**
**Systemic complications**	0.804	0.422–1.534	0.509			
**GCS**	**0.879**	**0.810–0.954**	**0.002**	**0.884**	**0.811–0.964**	**0.005**
**Hospital stay**	**0.844**	**0.808–0.882**	**0.000**	**0.836**	**0.793–0.882**	**0.000**
**ICU stay**	**0.943**	**0.914–0.972**	**0.000**	**0.899**	**0.855–0.946**	**0.000**
**Duration of ventilation**	0.999	0.998–1.000	0.063			
**Height (cm)**	**0.968**	**0.943–0.994**	**0.015**	1.006	00.976–1.037	0.698
**BMI**	0.989	0.933–1.048	0.711			
**Weight (kg)**	0.987	0.972–1.003	0.123			

SMRA: skeletal muscle radiation attenuation; VATI: visceral adipose tissue index; SMI: skeletal muscle index; SATI: subcutaneous adipose tissue index; ISS: Injury Severity Score; CCI: Comprehensive Complication Index, GCS: Glasgow Coma Score; ICU: intensive care unit; BMI: body mass index. Significant results are presented bold.

## Data Availability

Due to privacy restrictions all data are stored at the researchers institution. Qualified researchers will be able to gain access via application at the corresponding author.

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
