# Peer review of "The Impact of Body Mass Composition on Outcome in Multiple Traumatized Patients—Results from the Fourth Thoracic and Third Lumbar Vertebrae: A Single-Center Retrospective Observational Study"

_jcm, 2023, doi:10.3390/jcm12072520_

Round 1

Reviewer 1 Report

I reviewed the article by Belger et al, entitled “The impact of body mass composition on Outcome in Multiple Traumatized Patients-Results from the Fourth Thoracic and Third Lumbar Vertebrae” submitted to Journal of Clinical Medicine (Manuscript ID: jcm-2149107). In this retrospective observational study, the authors mainly assessed the relationship between body mass composition on CT and survival outcome and surrogate outcomes in patients with severe trauma. They found that a lower muscle mass and a high adipose tissue index are associated with a poor outcome in injured patients with injury severity scale >15. From this observation, they claimed the importance of assessing body mass composition on CT in trauma patients. First, the reviewer respects for the Authors' tremendous effort and time spent on this manuscript. However, there are numerous concerns with the data presentation, study design, as well as the statistical analysis. My concerns are listed below:

1

Title

Indicate the study’s design with a commonly used term in the title. Suggested title is "The impact of Body Mass Composition on Outcome in Multiple Traumatized Patients - Results from the Fourth Thoracic and Third Lumbar Vertebrae: A single center retrospective observational study".

Introduction

2

At the end of the introduction section, the authors state that the one object of this study was to clarify whether the body mass composition on a thoracic level provides comparable information to those gained from the validated lumbar level with regard to risk stratification in injured patients. Currently, this reviewer failed to understand the clinical implication for this object. Why body mass composition on a thoracic level is required? What is the clinical significance for it? Please clarfy.

3

If this study has any prespecified hypotheses, please describe it in introduction section.

Methods

4

Please present key elements of study design (e.g., A single center retrospective observational study) early in the paper

5

This observational study does not follow STROBE Statement Guidelines (https://www.equator-network.org/reporting-guidelines/strobe/). The authors should respect the basic rule of scientific writing.

6

Please describe settings and locations more in details (e.g. a tertiary hospital, academic hospital, referral trauma center, number of hospital and ICU beds, number of trauma surgery, and number of annual trauma admission, etc.) where the data were collected. This information should help readers to depict the context of this study more accurately. This reviewer thinks annual trauma volume is especially important because it can greatly affect the quality of trauma care [1, 2], and may be confounding factors in the analyses.

7

The ethical statement is missing. Manuscripts reporting studies involving human participants must include a statement on ethics approval and consent. Please include the name of the ethics committee that approved the study and the committee’s reference number and the relevant dates of approval. The author states that Ethics Approval is not applicable (page 16, line 399). The reviewer thinks this is absolutely wrong and not acceptable in the current ethical standard.

8

Who planned this study, who collected data, and who conducted the statistical analysis? I think if the same researchers are involved in study planning, data collecting, outcome measurement, and statistical analysis, there is a theoretical risk of biased assessment. Describe any efforts to address potential sources of bias. How to manage the quality of data?

9

Sample size calculation is missing. Explain how the study size was arrived at. If sample size was not determined a priori, please state so and provide post-hoc sample size estimation to provide the estimation of how was the power of this study.

10

Explain how missing data were addressed.

Results

11

Figure 1 and Figure 2

What can readers conclude from this data? This data might facilitate the reader's image but not informative. Such information should be moved to supplementary information file.

12

Several parameters were used without definitions. Please clearly define the skeletal muscle radiation attenuation (SMRA); skeletal muscle index (SMI); visceral adipose tissue index (VATI); and subcutaneous adipose tissue index (SATI) with appropriate references.

Table 1

13

Many vital information is missing. Give characteristics of study participants such as vital sings including GCS score, blood pressure, and respiratory rate, need of endotracheal intubation and mechanical ventilation, trauma etiology (penetrating or blunt), Abbreviated Injury Scale of each body parts, TRISS based probability of survival etc. At this current form, many readers including myself find it difficult to image the characteristics of study subjects. In addition, these variables would have confounded the results. There are too many unmeasured confounders. This is the serious flaw of your manuscript. The authors should adjust for such important confounders to provide more reliable data.

14

Definitions of type of injury is unclear.

15

Charlson Comorbidity Index (CCI) is a morbidity score that ranges from 1 to 6 points. However, according to table 1, mean CCI is 51.0178. This is absolutely unreliable. Such mistakes distract the reliability of data.

Table 3-7

16

There are too many statistical comparisons. Increased number of statistical comparisons greatly increase the risk of alpha errors. Therefore, this reviewer questions the quality of these data.

17

How did you choose each independent variables of multivariable model?

Discussion

18

Much of the discussion section is simply the list of previous studies, restate or rephrase the results and background that have already described in results and introduction section. Most parts of discussion are too speculative, and not based on the data obtained in this study. This reviewer thinks the discussion is not thought evoking one. In addition, the discussion section should indicate how the findings of this study can be used to solve the current problems. How and for what do we use these results presented here to improve the current trauma care and why?

18

The limitation section requires substantial revision. Please discuss limitations of the study, taking into account sources of potential bias or imprecision. Consider the important limitations and do not just list them but consider their relevance and how they might bias the results. Discuss both direction and magnitude of any potential bias.

19

Discuss the generalizability (external validity) and Implications for practice of the study results. Give a cautious overall interpretation of results considering objectives, limitations, multiplicity of analyses, results from similar studies, and other relevant evidence. This reviewer thinks that the generalizability of this study is limited, because this study was single center retrospective observational study.

20

Please indicate future research direction, immediately after limitation section.

Abbreviations

21

Many abbreviations are used in this manuscript. Keep abbreviations to a minimum. Please do not use non-standard abbreviations unless they appear at least three times in the text. In addition, Some abbreviations (e.g ISS) were used without definition. Please correct.

22

This manuscript does not conform the journal guideline. This is particularly true in the reference section which should be amended. This manuscript also contains a large number of careless mistakes. Please correct.

23

Give the source of funding and the role of the funders for the present study.

References

1. MacKenzie EJ, Rivara FP, Jurkovich GJ, Nathens AB, Frey KP, Egleston BL, Salkever DS, Scharfstein DO.A national evaluation of the effect of trauma-center care on mortality.N Engl J Med. 2006,26;354:366-78.

2. Minei JP, Fabian TC, Guffey DM, Newgard CD, Bulger EM, Brasel KJ, Sperry JL, MacDonald RD. Increased trauma center volume is associated with improved survival after severe injury: results of a Resuscitation Outcomes Consortium study. Ann Surg. 2014, 260:456-64.

Reviewer 2 Report

Well put together paper. No specific recommendations for this paper. Just general advice for making this paper better.

1) It is occasionally repetitive with ideas and this should be eliminated, Repetitiveness happens within sections of the paper and between sections of the paper. The paper is long enough and intense enough to read that it should be shortened where possible to make it more acceptable to the average reader.

2) Tables are difficult to read and very numerous. Would it be better to put some tables in another place like an appendix? They seem hard to read as well with the emboldening of numbers not entirely clear to the observer. This should be made more clear to the reader. Maybe just keep the minimum number of tables in the main paper and have the rest in an appendix.

3) The premise of this paper is not difficult to understand but the proof of this premise is very long and difficult to extract from the paper. It should be made simpler to read and less technical for the average reader. Again, maybe extra information appended in an appendix.

Overall, a strong paper where this reviewer cannot find fault. It is just a real chore to read this paper so it would be nice to simplify for the readers of this journal.

Reviewer 3 Report

Thank you for submitting this article on an interesting topic.

The authors provide data about prognostic relevance of body composition measurement on outcome after multiple trauma, and show comparable results of measurements on the level of TH4 and L3. This finding is of clinical relevance because it also allows prediction of the likelihood of complications in patients in whom CT scans were performed only in the thoracic region. I would like to report on some parts of the study where I think it still needs revision:

The introduction is very informative and covers the current literature and why this topic is worth looking into.

·         Line 66/67: Thus, the main goals of the present study were to a) determine whether there is a correlation between CT-based BC indices and a)

Materials and Methods is well structured.

·         Line 78: ICU, abbreviations should be written out when first mentioned (done at second time in Line 84)

·         Line 112: when you mention CD, it should say “CD classification” or “CD score”

·         Line 118: did you do an intra rater reliability assessment?

Results: Aside of some suggestions for improvement, this section is fine:

·         Line 164: there is a dot missing between sentences

·         Table 1: I would suggest to split the table into 3 parts: demographic data, typ of injury and ethiology (possibly like shown below) and clinical course. All this information in one table can not be brought into an acceptable form

·         Table 1: at which time was the mentioned RR measured?

·         Line 176: no need for new explanation of abbreviations

·         Tables 3-6: In my document these tables are deleted. However, you reference these tables in the following text.

·         Table 7: High should be Height

Discussion: very well written and of adequate length

Conclusion: clear statement supported by the data

However, due to journal standards an approval of an ethics committee is needed. I encourage the authors to obtain such an approval.

Round 2

Reviewer 1 Report

The authors have clarified several of the questions I raised in my previous review. Unfortunately, most of the major problems have not been addressed by this revision. As I stated in my previous review, I deem it unlikely that all those issues can be solved merely by a few added paragraphs. Instead there are still some fundamental concerns with the experimental design and, most critically, with the analysis. This means the strong conclusions put forward by this manuscript are not warranted and I cannot approve the manuscript in this form.

The reviewer cannnot understnad the clinical implication Why body mass composition on a thoracic level is required. Sience the patients multiple trauma patients require whole body CT, this reviewer thinks the author's response #2 is not correct.

The annual trauma volume is not provided.

Ethical approve is not provided.

Characteristics of study participants such as vital sings including GCS score, blood pressure, and respiratory rate, need of endotracheal intubation and mechanical ventilation, trauma etiology (penetrating or blunt), TRISS based probability of survival etc. At this current form, many readers including myself find it difficult to image the characteristics of study subjects. 

Discussion section is not revised satisfactory.

There are some other unsloved problems.

Studies submitted to JCM must be scientifically valid; for research articles, this includes a scientifically sound research question, the use of suitable methods and analysis, and following community-agreed standards relevant to the research field. This reviewer does not find that your study fulfills this criteria. 

This reviewer cannot be more positive on this occasion.

Author Response

Dear reviewer,

Thank you very much for the time you invested to review our manuscript. 

Yours sincerely

Esref Belger and Klemens Horst

Reviewer 2 Report

Good revision; no deficiencies noted.

Author Response

(The authors gave the same response as above.)

Reviewer 3 Report

Thank you for the correction of the points highlighted in the last review round. All of my points were adressed in a satisfactory manner.

I also appreciate the attempt to obtain permission from the ethics committee, however, this should have been done prior to evaluating the data. But the decision wether to accept this situation with a retrospective review board approval or not lies with the Editors.